# Rapid determination of solid-state diffusion coefficients in Li-based batteries via intermittent current interruption method

Yu-Chuan Chien [1,3], Haidong Liu[1], Ashok S. Menon [1,4], William R. Brant [1], Daniel Brandell [1] ✉ & Matthew J. Lacey [2] ✉

The galvanostatic intermittent titration technique (GITT) is considered the go-to method for determining the Li⁺ diffusion coefficients in insertion electrode materials. However, GITT-based methods are either time-consuming, prone to analysis pitfalls or require sophisticated interpretation models. Here, we propose the intermittent current interruption (ICI) method as a reliable, accurate and faster alternative to GITT-based methods. Using Fick's laws, we prove that the ICI method renders the same information as the GITT within a certain duration of time since the current interruption. Via experimental measurements, we also demonstrate that the results from ICI and GITT methods match where the assumption of semi-infinite diffusion applies. Moreover, the benefit of the non-disruptive ICI method to operando materials characterization is exhibited by correlating the continuously monitored diffusion coefficient of Li⁺ in a $LiNi_{0.8}Mn_{0.1}Co_{0.1}O_2$-based electrode to its structural changes captured by operando X-ray diffraction measurements.

As the demand for electrochemical energy storage surges, the research, development and application of new systems require comprehensive understanding of the electrochemical properties at an ever-increasing pace. A critical parameter for the community, from materials chemists to application engineers, is the diffusion coefficient of the charge carriers; i.e., Li⁺, in the case of Li-ion batteries. The galvanostatic intermittent titration technique (GITT) has been the most widely applied method for deriving the diffusion coefficient from electrochemical measurements. Derived from Fick's second law, GITT was first demonstrated in 1977 for a bulk $Li_3Sb$ electrode[1]. The technique consists of two repeating steps. First, a constant current is applied for a duration where the assumption of semi-infinite diffusion holds. Second, the current is switched off until the voltage becomes invariant, which indicates that equilibrium is reached. Through the analysis of the electrode potential measured during the current pulse and the change in the equilibrium potential, GITT renders the chemical diffusion coefficient of the charge-carrying ions. In the original GITT manuscript[1], the authors elaborated on the difference between chemical and tracer diffusion coefficients. Here, for simplicity, the former will be referred to as the diffusion coefficient in the following text. Later, the technique was applied to porous composite electrodes of Li-ion-insertion materials[2], which is the format of the majority of electrodes in state-of-the-art Li-ion batteries. Despite practical issues due to the geometry of composite electrodes[3], such as nonuniform current distribution, the technique serves as a powerful tool for the determination of diffusion coefficients if proper experimental parameters are chosen[4,5], e.g., appropriate current and duration of the current pulses. In addition, more sophisticated methods of extracting the experimental parameters enhances the accuracy and applicability of the GITT, for example, selecting only the portion of the voltage response governed by semi-infinite diffusion, using linear regression to derive its slope and fitting the voltage response to more elaborate non-Fickian diffusion models[4-6].

However, the time required to perform a GITT measurement remains one major drawback. In order to reach the equilibrium condition, the test cell has to be relaxed substantially longer than the time

[1]Department of Chemistry—Ångström Laboratory, Uppsala University, Box 538, Lägerhyddsvägen 1, 751 21 Uppsala, Sweden. [2]Scania CV AB, 151 87 Södertälje, Sweden. [3]Present address: Breathe Battery Technologies, Office 7, 35-37 Ludgate Hill, London EC4M 7JN, UK. [4]Present address: WMG, University of Warwick, Coventry CV4 7AL, UK. ✉e-mail: daniel.brandell@kemi.uu.se; matthew.lacey@scania.com

spent on applying current[5]. This results in an experiment that can be anywhere from 8 to 100 times longer than a typical galvanostatic test cycle[4,5]. Although the test may be accelerated by increasing the duration of current pulses and selecting only the initial data points for the analysis[7] or introducing a constant potential step instead of the relaxation step[8], these methods either reduce the number of measurements of diffusion coefficient or still require substantial amount of time. In addition, such long relaxation times make it difficult to couple GITT with simultaneous materials characterization with time constraints, e.g., diffraction or spectroscopy at synchrotrons and neutron sources, which can provide valuable structural and/or chemical information at the moment where the process under investigation takes place. Although recent studies suggest that the relaxation time can be decreased by nonlinear fitting of the voltage response, the advanced regression method may make the technique less accessible to the wider materials chemistry community[6].

In this work, an efficient, simple and non-disruptive alternative to the GITT is proposed: the intermittent current interruption (ICI) method[9–13]. The method introduces repeating transient current interruptions (usually 1 to 10 s) while the cell is under constant-current cycling. Through linear regressions of the potential change against the square root of step time during the current pauses, the time-independent and time-dependent parts of the resistance can be derived, which are termed internal resistance and diffusion resistance coefficient, respectively[11]. With the porous electrode model[14], it has been shown that the derived diffusion resistance coefficient is proportional to the coefficient of the Warburg element used when fitting electrochemical impedance spectroscopy (EIS) measurements[11]. Since the Warburg element describes both the capacitive behavior in porous electrodes[14,15] and diffusion processes[16–18], it is a logical consequence that the ICI method can also characterize diffusion processes in an electrochemical system.

Moreover, we show that the ICI method can be further developed to derive the diffusion coefficient through simple data analysis in less than 15% of the experimental time of GITT. The theoretical derivation confirms that the voltage response in the ICI method is analogous to that in the GITT given that the datapoints are within a limited time interval since the current interruption. An experimental example with $LiNi_{0.8}Mn_{0.1}Co_{0.1}O_2$ (NMC811) as the working electrode corroborates the equivalence of the GITT, ICI and EIS methods for characterizing $Li^+$ diffusion. Finally, the combination of operando X-ray diffraction (XRD) and the ICI method is demonstrated to directly correlate the structural evolution to the Li-ion mobility. This example manifests not only the efficiency of the ICI method in probing the transport properties, but also its compatibility with operando techniques. Moreover, it also indicates the potential of the automatable ICI method as a tool for state of heath estimation of Li-ion batteries.

## Results and discussion

### Brief summary of the intermittent current interruption method

The intermittent current interruption (ICI) method was originally designed for continuous resistance measurements. During constant-current cycling of a diffusion-controlled system, the method introduces transient current pauses, in which the change in electrode potential ($\Delta E$) and time ($\Delta t$) since the current ($I$) is switched off can be expressed as Eq. 1[10,11,19], given $\Delta t$ is sufficiently small, as elaborated in Supplementary Note 1.

$$\Delta E(\Delta t) = E(\Delta t) - E_I = -IR - Ik\sqrt{\Delta t} \qquad (1)$$

where $E_I$ is the potential right before the current is switched off, and $R$ and $k$ are termed internal resistance and diffusion resistance coefficient, respectively, which are derived by extracting the intercept and slope by linear regression of $\Delta E$ against $\sqrt{\Delta t}$ automatically with a script in the R programming language[20]. In this work, the method will be

further developed to derive the diffusion coefficient of the charge carrier in insertion-type electrode materials, which is otherwise mostly done using the GITT in the literature[1,2,4,5,7,21–23].

The applications of both the GITT and the ICI method are illustrated by parts of the raw data of the same cell in different cycles in Fig. 1. On the working electrode of NMC811, the ICI method introduces short pauses (10 s every 300 s) during constant-current (C/10, C = 200 mA g$^{-1}$) charging while the GITT applies short current pulses (C/10, 600 s) in between long rests (>1 h) for reaching the open-circuit potential (OCP). Therefore, the ICI method probes the same range of states of charge in less than 15% of the time required by the GITT.

The efficiency of the ICI method is brought by the new approach to acquire the experimental parameters for deriving the diffusion coefficient, which is elaborated in the Methods section and summarized here. Both the GITT and ICI method render the diffusion coefficient ($D$) using the following equation.

$$D = \frac{4}{\pi}\left(\frac{V}{A}\frac{\frac{\Delta E_{OC}}{\Delta t_I}}{\frac{dE}{d\sqrt{t}}}\right)^2 \qquad (2)$$

where $V$ is the molar volume of the electrode materials, $A$ is the surface area of the electrode, $E_{OC}$ is the OCP, $\Delta t_I$ is the period when a constant current is applied between consecutive OCP measurements, $E$ is the electrode potential and $t$ is the step time, which refers to the "current pulse" step for the GITT and "current pause" step for the ICI method, respectively. In addition to extracting $dE/d\sqrt{t}$ during current pauses, instead of current pulses, the ICI method circumvents the long rest for the open-circuit condition by approximating the OCP slope ($\Delta E_{OC}/\Delta t_I$) by the slope of the iR-corrected pseudo-OCP. To further improve the approximation of the OCP by the pseudo-OCP measured at a low C-rate (C/10 in this work), with the iR-drop that is obtained frequently by the ICI method, the iR-corrected pseudo-OCP can be calculated, of which the slope is very close to the OCP slope obtained after long rests. Both of these new approaches will be examined by the following experimental results.

### Comparison between the results from the GITT and ICI method

A modified GITT protocol is designed to compare the results from the GITT, the ICI method as well as EIS measurements employed as a reference, as depicted in Fig. 2. Two identical three-electrode non-aqueous Li metal cells with NMC811 as the working electrode were assembled, which showed similar behaviors in the two cycles of the modified GITT protocol. Thus, the results of first cell (Cell 1) in the first cycle are discussed in the main text while the rest (the second cycle of Cell 1 and both cycles of Cell 2) are presented in the Supplementary Figs. 1 to 7.

As mentioned above and elaborated in the Methods section, to derive the diffusion coefficient, two measurements are required: $dE/d\sqrt{t}$ during semi-infinite diffusion and the OCP slope. Therefore, the following text will first compare the two values obtained by the GITT and ICI method. Then, the diffusion coefficients calculated from the two methods will be presented. The data acquired during the current pulses and the rest periods are analyzed by the GITT and ICI methods, respectively, as indicated in Fig. 2. For the GITT, two data selection intervals, 5–40 and 50–150 s, were utilized because they contain the linear region of the $E - \sqrt{t}$ plot above and below 3.7 V, respectively, as elaborated in Supplementary Note 2. An example of each case is plotted in Supplementary Fig. 8. Since more than 75% of the capacity of NMC811 is above 3.7 V, the GITT results in the following text are for clarity derived from the 5–40 s interval while results from both intervals can be found for both cells in Supplementary Figs. 1 to 7. For the ICI method, the interval was chosen to be 1–5 s because it contains the linear region of the $E - \sqrt{t}$ plot during current pauses, as shown in Supplementary Fig. 8c.

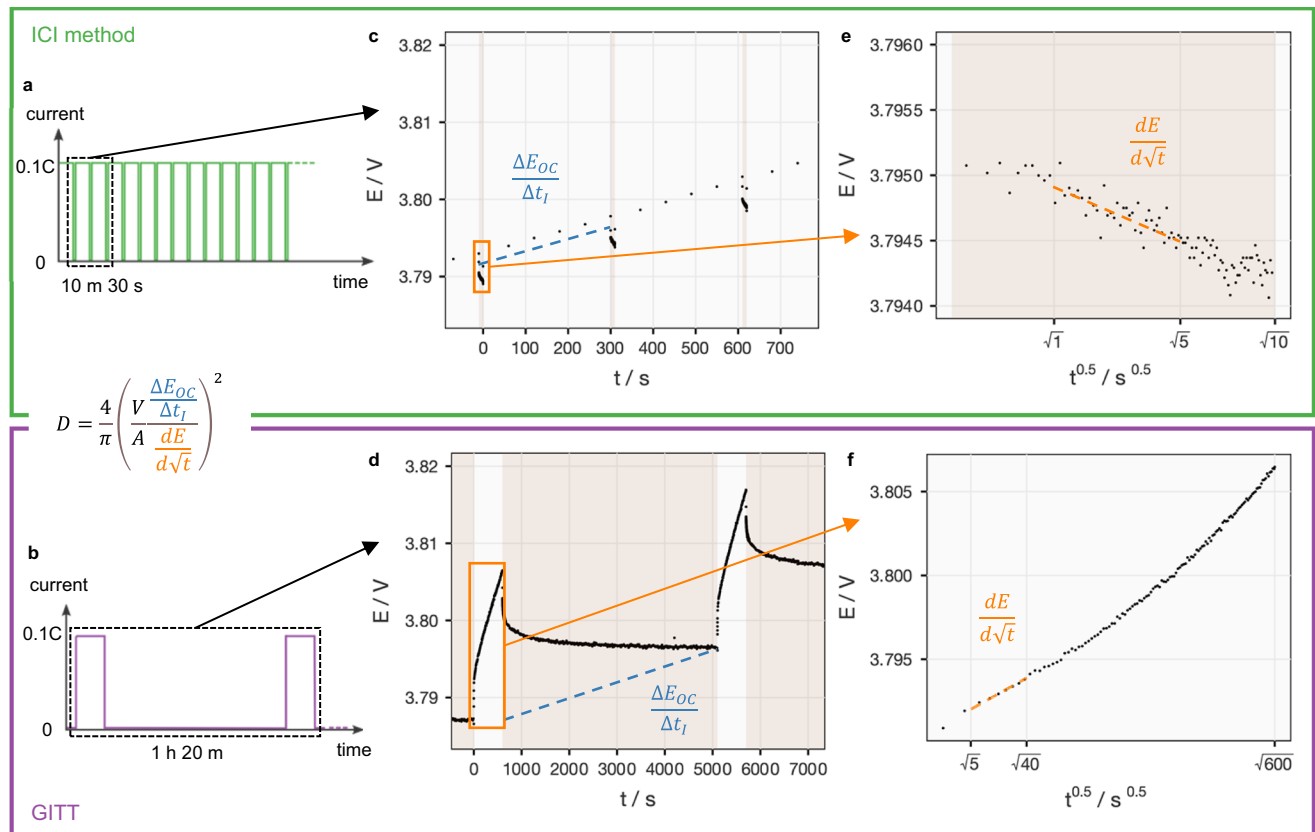

**Fig. 1 | Comparison between the ICI and the GITT methods. a, b** How the current is controlled in the ICI method and the GITT in this work on the same test time scale. **c, d** Electrode potential ($E$) versus step time ($t$) as a response from the current input in the dashed-line box in a and b, respectively. The blue dashed line indicates where the slope of open-circuit potential against current-passing time ($\Delta E_{OC}/\Delta t_I$) is obtained. **e, f** Electrode potential ($E$) plotted against the square root of step time ($t^{0.5}$) with datapoints in the orange box in **c**, **d**, respectively. The orange line shows how the derivative ($dE/d\sqrt{t}$) is obtained. In **c–f** plots, the white background denotes the duration where a 0.1 C current is applied while the brown-shaded intervals are where no current is applied. Please refer to the explanation of Eq. 2 for the symbols used in the shared equation for both methods.

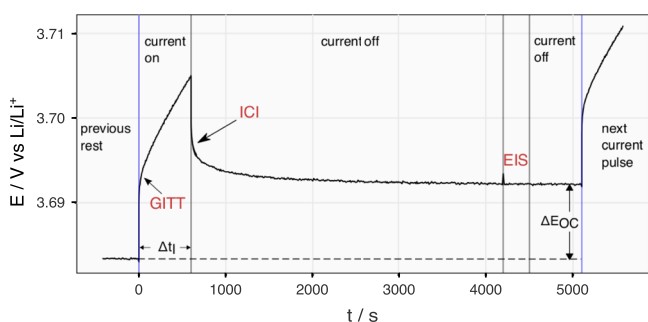

**Fig. 2 | Modified GITT protocol designed to compare the results from the GITT, ICI method and EIS.** Electrode potential ($E$) plotted against time ($t$) during a 'current on' step ($I = 20\ mA\ g^{-1}$) and a 'current off' step ($I = 0$) of the modified GITT program used in this work for the comparison of the results from the GITT, ICI method and EIS. Note that the long rest period is not necessary for the ICI method. It is done in this manner here for the comparison between the three methods.

Figure 3 displays the $k$ values, which are $dE/d\sqrt{t}$ normalized by the current (Eq. 1) from GITT and ICI, and the Warburg coefficients ($\sigma$) multiplied by $\sqrt{8/\pi}$ from EIS fittings. This linear relationship of $k = \sigma\sqrt{8/\pi}$ has been demonstrated in previous work[11]. Above 3.8 V, the $k$ values determined by ICI and GITT are close to each other, which confirms the theoretical derivation in "Summary of the derivation of the GITT and the ICI method" paragraph in the "Methods" section and is corroborated by the EIS results. During discharge, the $k$ values obtained from EIS are slightly higher than those from the GITT

and ICI measurements, while the latter two remain close to each other. In Supplementary Fig. 9, a statistical analysis on the relative difference between the $k$ values from the GITT and ICI methods is exhibited. The close-to-zero average indicates that no systematic error appears, while the standard deviation being at 26% is promising when considering the large range of the GITT measurements of this same material in the literature, which scatters over two orders of magnitude[24].

Below 3.7 V, the linear region on the $E - \sqrt{t}$ plots of the GITT measurements shifts to 50–150 s, as demonstrated in Supplementary Fig. 8, which indicates an increase in the time constant of the charge-transfer related process. This phenomenon is also manifested by the expansion of the second semi-circle on impedance spectra (Supplementary Fig. 10 and Supplementary Note 3), which do not show a Warburg element within the frequency range (20 kHz–10 mHz) and thus cannot render $k$ values below 3.7 V. The increase in charge transfer resistance represented by the enlarged semi-circle has been reported for NMC111 ($LiNi_{0.33}Mn_{0.33}Co_{0.33}O_2$) at low state of charge (SoC), which pushes the Warburg element to frequencies lower than 10 mHz[2,18]. This indicates that the EIS above 10 mHz, GITT with 5–40 s data selection interval and ICI method do not characterize the diffusion process below 3.7 V. Between 3.7 and 3.8 V, a mismatch between the GITT and ICI methods can be observed during charging, but not during discharging. The difference can be an effect of the varying impedance in the course of charging and discharging[5,25] and different data selection time intervals of the two techniques. In summary, Fig. 3 illustrates the consistency of EIS, GITT and ICI method within their respective limitations.

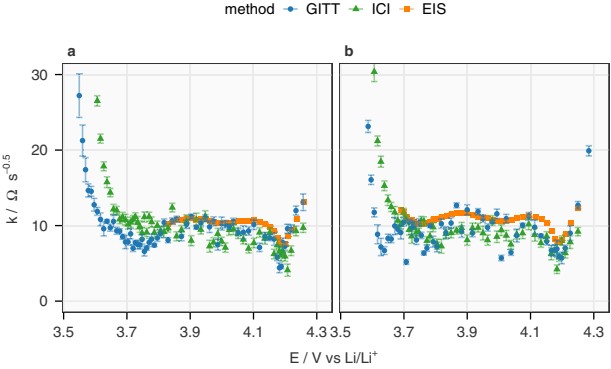

**Fig. 3 | Comparison between the diffusion resistance coefficient derived from the GITT, ICI method and EIS.** The diffusion resistance coefficient ($k$) in NMC811 in Cell 1 during charging and discharging are plotted against the OCP of the electrode ($E$) in **a**, **b**, respectively. The GITT data were fitted with data selection interval 5–40 s. The EIS results are derived from the linear relationship between k and the coefficient of the Warburg element ($\sigma$, k = $\sigma\sqrt{8/\pi}$). The maximum of the $y$-axis is set to 30 $\Omega$ s$^{-0.5}$ to show the differences of the data above 3.7 V. Due to a technical issue, the spectra below 3.8 V in the first charge were not properly collected, but it was solved afterwards. The analysis of the difference between the values derived from the GITT and ICI methods can be found in Supplementary Fig. 11. The error bars originate from the standard deviation of linear regression.

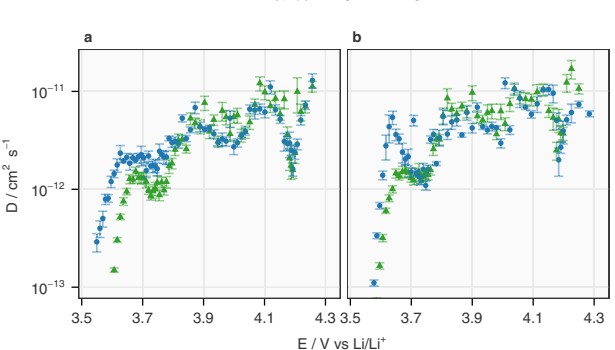

**Fig. 5 | Comparison between the Li-ion diffusion coefficient derived from the GITT and ICI method.** The Li-ion diffusion coefficient in NMC811 ($D$) in Cell 1 at various OCP of the electrode ($E$) derived from the GITT with data selection interval 5–40 s and the ICI method during charging and discharging are plotted in **a**, **b**, respectively. The minimum of the $y$-axis is set to 10$^{-13}$ cm$^2$ s$^{-1}$ to show the differences of the data above 3.7 V. The analysis of the difference between the values derived from the GITT and ICI methods can be found in Supplementary Fig. 5. The error bars originate from the errors in k, as shown in Fig. 3, which stem from the standard deviation of linear regression.

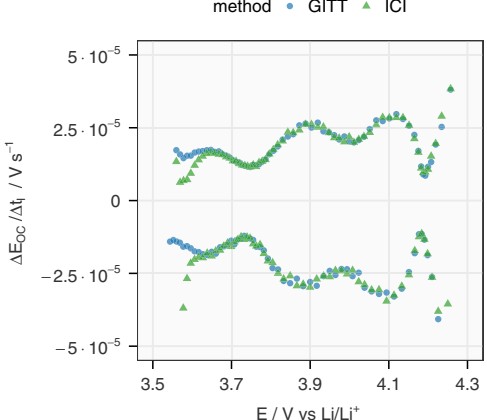

**Fig. 4 | Comparison between the slope of OCP derived from the GITT and ICI method.** The slope of the OCP ($\Delta E_{OC}/\Delta t_I$) obtained from the relaxed potentials at the end of each rest period (GITT) is compared with the slope of the potential under constant-current load subtracting the iR-drop derived from the ICI method ($\Delta[E(\Delta t = 0)]/\Delta t_I$, marked as ICI). The values are obtained from Cell 1 in the first cycles and the positive and negative during charging and discharging, respectively. The analysis of the difference between the values derived from the GITT and ICI methods can be found in Supplementary Fig. 12.

The other quantity experimentally determined in these methods used in the calculation of the diffusion coefficient (Eq. 2) is the OCP slope. Figure 4 presents a comparison between the slopes of the OCP obtained from the relaxed potential at the end of the rest period in the GITT protocol and the iR-corrected pseudo-OCP in the ICI analysis. The difference between the values from the two methods are minimal above 3.65 V, which is confirmed by the close-to-zero average and minimal standard deviation (0.086) of the relative difference in Supplementary Fig. 11. The deviation at low SoC is presumably linked to the high charge transfer resistance discussed above, which increases the time-constant of electrochemical double-layer charging and thus interferes with the resistance determination of the ICI method. Nevertheless, the good agreement between the slopes of OCP and iR-corrected pseudo-OCP in most SoC intervals indicates that the ICI

method alone can deliver the required electrochemical parameters for the calculation of the diffusion coefficient. By skipping the time-consuming relaxation periods, the ICI method can save around 90% of the time spent on common GITT protocols, such as the one used in this work.

With both experimental inputs verified, the diffusion coefficients of Li$^+$ in NMC811 at various SoC obtained by the GITT and ICI method are exhibited in Fig. 5. Overall, the results from the three analyses are close to each other and previously reported Li$^+$ diffusion coefficients in NMC811[5,26]. Above 3.8 V, the match is close for the values from the GITT and the ICI method. This is expected since the $dE/d\sqrt{t}$ values derived from both GITT and ICI method are in good agreement above 3.8 V in Fig. 3 and the slopes of OCP from both methods are basically the same above 3.65 V in Fig. 4. Below 3.7 V, differences between the two analyses are obvious since neither method characterizes the diffusion process, as discussed above. Between 3.7 and 3.8 V, the match during discharge and the slight mismatch during charge are also expected from the comparison in Fig. 3. Including the results from Cell 2 in Supplementary Fig. 5 and the statistical analysis on the relative difference between the diffusion coefficients from the two methods in Supplementary Fig. 12, it can be concluded that both methods generally agree with each other above 3.7 V, while a more consistent match can be found above 3.8 V. This demonstrates the validity of the ICI method as an efficient alternative to GITT.

Other valuable information provided by the ICI method is the internal resistance ($R$), as shown in Fig. 6. $R$ values derived from the iR-drop in the GITT and ICI method are compared with the sum of R0, R1 and R2 from the fitting results of EIS, of which the equivalent circuit model is displayed in Supplementary Fig. 13. The results from ICI and EIS are almost identical across the whole range of SoC. The GITT yields similar $R$ values during discharge but larger values upon charging. Nonetheless, all methods confirm the high internal resistance below 3.7 V, which changes the linear region of $E - \sqrt{t}$ plots, as discussed above. The internal resistance has been reported to be an important indicator for ageing of NMC811[27] and utilized for detecting Li-plating in commercial Li-ion cells[28].

**Continuous measurements of diffusion coefficient and internal resistance using the ICI method**

The value of the ICI method's efficiency is illustrated by the results in Fig. 7, which show the change in Li-ion diffusion coefficient and

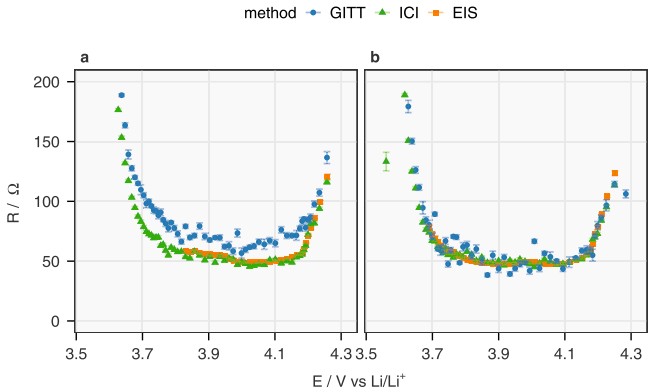

**Fig. 6 | Comparison between the internal resistance derived from the GITT, ICI method and EIS.** The internal resistance ($R$) of NMC811 in Cell 1 at various OCP of the electrode ($E$) derived from the GITT with data selection interval 5–40 s, the ICI method and the EIS fitting (R0 + R1 + R2 in the equivalent circuit model in Supplementary Fig. 13) during charging and discharging are plotted in **a**, **b**, respectively. The maximum of $y$-axis is set to 200 Ω to show the differences of the data above 3.7 V. The error bars originate from the standard deviation of the linear regression.

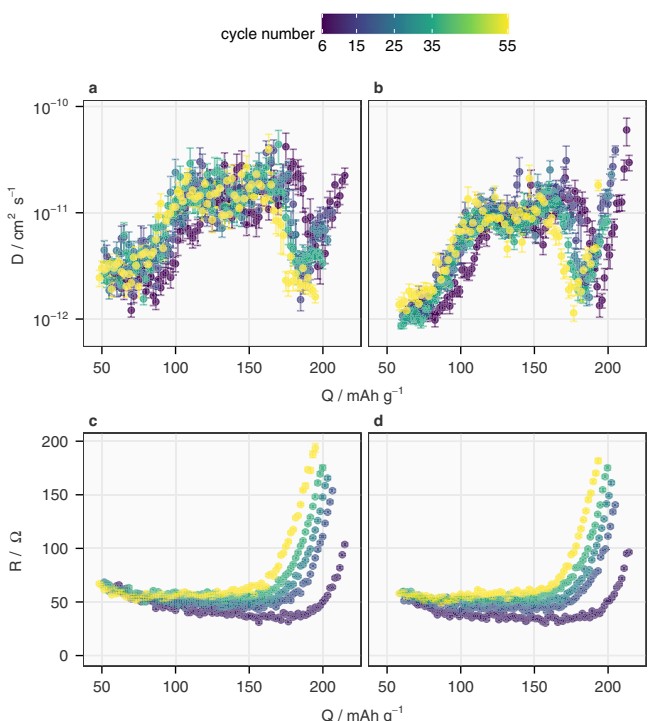

**Fig. 7 | Continuous measurements of diffusion coefficient and internal resistance using the ICI method during long-term cycling.** In cycles 6, 15, 25, 35 and 55, the Li-ion diffusion coefficient in NMC811 ($D$) of Cell 1 derived by the ICI method is plotted against the specific capacity ($Q$) during charging and discharging are plotted in **a**, **b**, respectively, while the internal resistance ($R$) from the same ICI measurements during charging and discharging are plotted in **c**, **d**, respectively. Only values with $E \geq 3.7$ V, where the ICI method is applicable as discussed above, are shown. The error bars originate from the standard deviation of linear regression.

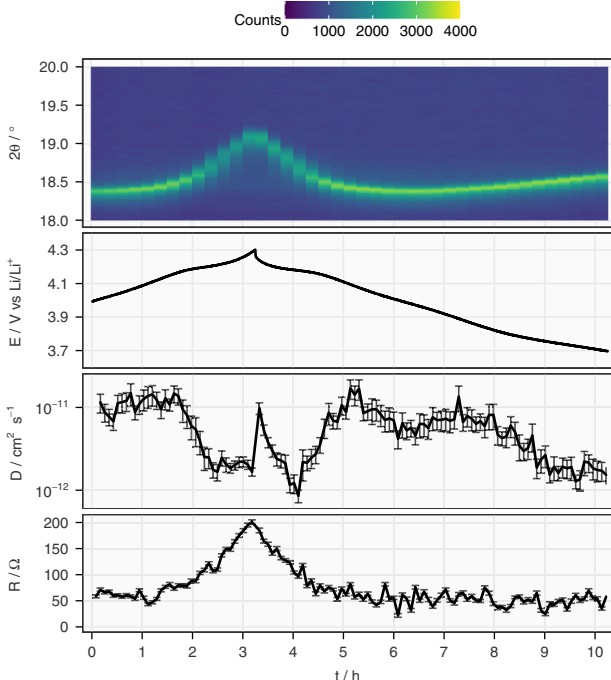

**Fig. 8 | Results from operando X-ray diffraction combined with the ICI method.** The evolution of the 003 (first panel) reflections (Cu-K$\alpha_1$) as a heat map, electrode potential (E), Li-ion diffusion coefficient in NMC811 ($D$) and the internal resistance (R) from the combination of operando XRD and the ICI method conducted on Cell 1 after 56 galvanostatic cycles between 3.0 and 4.3 V and the 57th charge to 4.0 V. The incomplete cycle does not affect the observation of the correlation between the Li-ion diffusivity and the structural change above 4.2 V. The error bars of D and R originate from the standard deviation of linear regression.

higher in the first 15 cycles than in the following cycles. This SoC range corresponds to the drastic shrinkage of the $c$ lattice parameter of the rhombohedral ($R\bar{3}m$) unit cell, which has been reported in previous operando XRD studies of NMC811[22,29–31]. To confirm the correlation between the decreasing Li-ion diffusivity observed here and the microstructural evolution of NMC811, a combination of operando X-ray diffraction (XRD) and the ICI method was carried out after the 56th galvanostatic cycle.

## Combination of operando XRD and the ICI method

The operando XRD experiment coupled with the ICI method was performed on Cell 1 after 56 cycles. As depicted in Fig. 8, when the 003 reflection ($R\bar{3}m$) shifts to higher 2θ values, the diffusion coefficient drops over an order of magnitude and the internal resistance tripled. The shift of the 003 reflection starts at 4.1 V and accelerates at 4.2 V, which coincides with the start of the increase in internal resistance and the decrease in diffusion coefficient, respectively. The reverse can be observed during discharge. In addition, by comparing the patterns taken above 4.2 V with a previous operando XRD study on the same material in the first cycle[30], it can be observed that in the degraded cell here, the 003 reflection is composed of two rhombohedral phases with dissimilar c lattice parameters (Supplementary Figs. 14 to 17). The exact mechanism for this phase separation is still debated, but most models attribute this to the ramifications of the formation of the degraded rock salt phase[29,32,33]. Nevertheless, it is shown here that the ICI method can be effectively combined with operando XRD and track the diffusion coefficient and internal resistance in real time, which constitutes a valuable method for further studies of the degradation mechanisms of battery materials.

In summary, this work establishes the theoretical foundation and experimental validation for the application of the ICI method as an

internal resistance over more than 50 cycles of constant-current charge and discharge. While the internal resistance increases more uniformly in all SoC, a clear decrease in the Li-ion diffusion coefficient can be observed above 4.2 V, which corresponds to -200 mAh g⁻¹ in cycle 6 and -180 mAh g⁻¹ in the rest of the cycles. The rate of decrease is

efficient alternative to GITT. Provided that 1) the diffusion process under investigation exhibits the semi-infinite diffusion behavior within the maximum time allowed for the current interruption and that 2) the pseudo-OCP slope is a good approximation of the true OCP slope, the ICI method can yield the diffusion coefficient with a much shorter experimental time. In the case of our validation experiment with NMC811, more than 85% of the time required for a typical GITT experiment can be saved. Moreover, the internal resistance and diffusion resistance coefficient (or equivalently, the Warburg coefficient) determined by the ICI method are also verified by EIS for NMC811. The efficient determination of diffusivity and resistance unlocks new applications which GITT and EIS are deemed too time- or resource-consuming, such as online cell parameterization for adaptive charging protocol and simultaneous observation of *operando* spectro-/diffractometry and electrochemical impedance/resistance. When exemplified by a combination of *operando* XRD and the ICI method, the rapid decrease of the Li-ion diffusion coefficient above 4.2 V over cycling could be correlated to the increasing irreversibility of the contraction and elongation of the *c* lattice parameter of the NMC structure.

## Methods

### Summary of the derivation of the GITT and the ICI method

The derivations of both galvanostatic intermittent titration technique (GITT) and the intermittent current interruption (ICI) method start from solving the equations describing Fick's second law[1,21,34–36]:

$$\frac{\partial C(r,t)}{\partial t} = D\frac{\partial^2 C(r,t)}{\partial r^2} \qquad (3)$$

where $C$ is the concentration of the diffusing species, $r$ is the radial distance in the spherical coordinates, $t$ is time and $D$ is the diffusion coefficient. The boundary conditions with an applied current $i(t)$ and an initial concentration $C_O$ are as follows:

$$\begin{cases} -D\frac{\partial C(r_p,t)}{\partial r} = \frac{i(t)}{nFA} \\ C(r,0) = C_0 \end{cases} \qquad (4)$$

where $r_p$ is the radius of the electrode particle, $n$ is the charge number of the diffusing species (which is 1 for Li⁺), $F$ is the Faraday constant and $A$ is the area of the surface where the diffusing ions enter the electrode material. For the GITT, a constant current $I$ is applied.

$$i(t) = I \qquad (5)$$

While the full solution is detailed in Eq. 3 in Supplementary Note 1, when $t << r_p^2/D$, semi-infinite diffusion can be assumed and the concentration at the surface $C_s$ can be expressed as:

$$C_s(t) = C(r_p,t) = C_0 - \frac{2I\sqrt{t}}{FA\sqrt{D\pi}} \qquad (6)$$

$$\frac{dC_s}{d\sqrt{t}} = -\frac{2I}{FA\sqrt{D\pi}} \qquad (7)$$

Suppose the change in concentration at the particle surface is small and thus linear to the change in the electrode potential $E$ (i.e., the potential increases linearly with decreasing Li concentration), the above expression can be expanded to:

$$\frac{dE}{d\sqrt{t}} = \frac{2I}{FA\sqrt{D\pi}}\frac{dE}{dC_s} \qquad (8)$$

With both derivatives of $E$ extracted from experimental data, the GITT renders the diffusion coefficient by the following relationship.

$$D = \frac{4}{\pi}\left(\frac{I}{FA}\frac{\frac{dE}{dC_s}}{\frac{dE}{d\sqrt{t}}}\right)^2 \qquad (9)$$

Instead of analyzing the potential change from open circuit to a constant current load, the ICI method utilizes the opposite case where the current is switched to zero from a constant current load. Considering a constant current $I$ being applied from $t = 0$ and switched off at $t = \tau_1 > 0$, Eq. 5 is changed to:

$$i(t) = I - IH(t) \qquad (10)$$

where $H$ is the Heaviside function, defined as:

$$H(t) = \begin{cases} 0, t < \tau_1 \\ 1, t \geq \tau_1 \end{cases} \qquad (11)$$

By inserting Eq. 10 into Eq. 4 as the boundary conditions, Eq. 3 can now be solved as the following by the Zero-Shift Theorem[16].

$$C_s(t) = C_0 + F(t) - H(t)F(t - \tau_1) \qquad (12)$$

where $C_0 + F(t)$ is the full solution of Eq. 3 when $i(t) = I$, shown in Eq. 3 in Supplementary Note 1. $F(t - \tau_1)$ can be approximated by the semi-infinite diffusion case ($t << L^2/D$) since the ICI method only analyzes the potential change in a short period $\Delta t$ after the current is switched off, which means $\Delta t = t - \tau_1 << L^2/D$. Thus, when $t \geq \tau_1$, Eq. 12 can be written as:

$$C_s(t) = C_0 + F(t) + \frac{2I\sqrt{t - \tau_1}}{FA\sqrt{D\pi}} \qquad (13)$$

So, the surface concentration at $t = \tau_1 + \Delta t$ is:

$$C_s(\tau_1 + \Delta t) = C_0 + F(\tau_1 + \Delta t) + \frac{2I\sqrt{\Delta t}}{FA\sqrt{D\pi}} \qquad (14)$$

Assuming that $\tau_1$ is so much larger than $\Delta t$ that $F(\tau_1 + \Delta t) \approx F(\tau_1)$ and thus independent of $\Delta t$, which is a criterion for the ICI method and is discussed in Supplementary Note 1, the following is obtained:

$$\frac{dC_s(\Delta t)}{d\sqrt{\Delta t}} = \frac{2I}{FA\sqrt{D\pi}} \qquad (15)$$

The above relationship between $C_s$ and $\Delta t$ is analogous to Eq. 7 because the current pause can be interpreted as the result of applying a current pulse of the same magnitude but in the opposite direction on top of the existing constant current. Inserting $-I$ as $I$ into Eq. 7 cancels out the negative sign on the right side of the equation and renders Eq. 15. If the change in concentration at the particle surface is small, the same assumption applied for Eq. 8, Eq. 15 can be expanded to:

$$-\frac{dE}{d\sqrt{\Delta t}} = \frac{2I}{FA\sqrt{D\pi}}\frac{dE}{dC_s} \qquad (16)$$

With both $dE/d\sqrt{\Delta t}$ and $dE/dC_s$ obtained from the ICI method, as demonstrated in the "Experimental execution of the ICI method" paragraph, the diffusion coefficient can then be calculated from Eq. 9 by substituting $t$ with $\Delta t$.

Since the ICI method is derived from the same theoretical origin as that of the GITT, the assumptions of the GITT are inherited by the ICI

method, such as that the solid state diffusion in the probed electrode dominates the time-dependent cell resistance and that the material is a single-phase solid solution.

## Experimental execution of GITT

To employ Eq. 9 in a conventional GITT measurement, the two derivatives of $E$ ($dE/dC$ and $dE/d\sqrt{t}$) have to be determined. For $dE/dC$, the change in concentration of the charge carrier is not directly measured but can be calculated under constant current.

$$dC = \frac{I dt_I}{FV} \tag{17}$$

where $V$ is the volume of the electrode and $dt_I$ is the duration of the applied current. Assuming that $dE/dC$ changes relatively slowly and thus can be interpolated by $\Delta E/\Delta C$[1,7], Eq. 9 can be rewritten as Eq. 2.

It is worth noting that Eq. 17 applies to the concentration of the entire electrode. In order to relate the measured electrode potential, which reflects the surface concentration, to the bulk concentration, the measurement should ideally be done when the concentration is uniform throughout the electrode. This is indicated by the fully relaxed electrode potential, i.e., $dE/dt = 0$, also known as the open-circuit condition. In practice, it takes long time to achieve equilibrium in the electrode, which is the reason for the substantial time consumption of GITT[5].

$dE/d\sqrt{t}$ is the slope on the plot of $E$ against $\sqrt{t}$. In the original GITT paper[1], Eq. 2 is further reduced by assuming that $E$ is linear to $\sqrt{t}$ during the whole current pulse. However, this assumption is less likely to hold for electrode particles in a composite electrode since the duration for the semi-infinite diffusion condition, $t \ll L^2/D$, is reduced by the shorter $L$, compared to the bulk electrode used in the original paper. Two solutions to solve the issue are: (1) selecting only the data lying in the linear region on the $E$-$\sqrt{t}$ plot or (2) fitting the data with the full solution (Eq. 3 in Supplementary Note 1) to Fick's second law (method P3 and P5 in the reference, respectively)[4]. In this work, the first solution is employed and the effect of data selection on the GITT analysis is examined in Supplementary Fig. 8.

## Experimental execution of the ICI method

Comparing Eqs. 1 and 16, it can be observed that

$$Ik = -\frac{dE}{d\sqrt{\Delta t}} = \frac{2I}{FA\sqrt{D\pi}}\frac{dE}{dC_s} \tag{18}$$

which can be reorganized into the form of Eq. 2 with Eq. 17.

$$D = \frac{4}{\pi}\left(\frac{V}{A}\frac{\frac{\Delta E_{OC}}{\Delta t_I}}{-\frac{dE}{d\sqrt{\Delta t}}}\right)^2 = \frac{4}{\pi}\left(\frac{V}{A}\frac{\frac{\Delta E_{OC}}{\Delta t_I}}{Ik}\right)^2 \tag{19}$$

The above equation is the same as Eq. 2 if $t$ in Eq. 2 is interpreted as step time of the step where $dE/dt$ is measured, which is when the current is on and off for the GITT and the ICI method, respectively.

Without the relaxation step in the ICI method, $\Delta E_{OC}/\Delta t_I$, can be approximated by the slope of iR-corrected pseudo-OCP. From two neighboring current interruptions, the change in OCP can be approximated by the change in $E(\Delta t = 0)$, which is the potential right before the current pause subtracting the iR-drop, as shown in Eq. 1. The validity of using $\Delta[E(\Delta t = 0)]/\Delta t_I$ as $dE_{OC}/dt_I$ is examined in "Results and discussion."

## Experimental method

Two identical three-electrode non-aqueous Li metal cells were assembled, which are referred to as Cell 1 and Cell 2. The working electrode (∅13 mm) was a tape-cast composite electrode consisting of 90 wt%

NMC811 powder (Customcells Itzehoe GmbH), 5 wt% of carbon black (Super C65, Imerys) and 5 wt% poly(vinylidene difluoride) (PVdF, Solvay)[30]. According to the supplier, the NMC811 particles have a median diameter of 4 μm and specific surface area of 1.5 m² g⁻¹, determined by the Brunauer–Emmett–Teller (BET) analysis of the nitrogen adsorption isotherm. The specific volume was calculated from the molecular mass and the previously reported (rhombohedral) unit cell parameters obtained through X-ray diffraction (XRD) to be 0.63056 cm³ g⁻¹ [30]. The areal loadings of NMC811 were 2.11 and 2.13 mg cm⁻² in Cell 1 and Cell 2, respectively. Both the counter (∅15 mm) and reference electrodes (ring with inner and outer diameters of 16 and 22 mm, respectively) were metallic lithium (China Energy Lithium Co., Ltd, 130 μm thick, purity: 99.9%). As sketched in Supplementary Fig. 18, the reference electrode was placed between the working and counter electrodes with separators (Celgard® 2325) on both sides according to a previously reported cell geometry[12]. The cells were assembled in an Ar-filled glove box (water and oxygen content below 1 ppm) with 105 μL electrolyte of 1 M LiPF₆ in a mixture of ethylene carbonate/diethylene carbonate (EC/DEC 1:1 by volume, Solvionic, purity: 99.9%, water content below 20 ppm) added before sealed in pouch bag material. Assembled cells were rested for 12 h before 3 pre-cycles at 20 mA g⁻¹ between 3.0 and 4.3 V. The specific current values in this work are calculated based on the mass of the active material in the working electrode, i.e., NMC811. Electrochemical tests were carried out using a Biologic MPG 2 in a laboratory air-conditioned to 22 °C. Although the cell temperature was not precisely controlled, its effect on the electrochemical techniques examined here should be equal.

A modified GITT protocol was designed here to compare the GITT, ICI and EIS at the same SoC, which is schematically shown in Fig. 2. A constant current of 20 mA g⁻¹ (corresponding to C/10, where C is here defined as 200 mA g⁻¹ for NMC811) was applied for 10 min, which was followed by a 1-h rest. The low specific current is chosen to minimize the concentration gradient in the electrolyte so that the diffusion resistance revealed by both the GITT and the ICI method is dominated by solid state diffusion. During the first minute of the rest, potential was recorded every 0.1 s for the ICI analysis. After the rest, an EIS measurement was performed from 20 kHz to 10 mHz with an amplitude of 10 mV. Another 10-min rest followed the EIS measurement before the next current pulse. The modified GITT protocol was applied between 3.0 and 4.3 V for two cycles. In the second discharge, the cutoff was lowered to 2.0 and 2.5 V for cell 1 and 2, respectively. For both GITT and ICI analysis, the electrode volume and area were approximated by the volume and surface area of the NMC particles stated above. The BET-surface area may differ from the electrochemically active surface area. However, the objective of this work is to compare the GITT and ICI method, both of which are equally affected by this factor. The impedance spectra, where a Warburg element is present, were fitted to the equivalent circuit model in Supplementary Fig. 13 by a modified Levenberg–Marquardt algorithm provided by the "minpack.lm" package in the R programming language[37].

After the above-described test, the cells went on to be cycled with the standard ICI protocol. Both cells were charged to 4.3 V and discharged to 3 V at a constant current of 20 mA g⁻¹. A 10-s current interruption every 5 and 15 min was introduced to Cells 1 and 2, respectively. During the 57th discharge of Cell 1, an operando XRD experiment was performed on the cell as it was charged up to 4.3 V and subsequently discharged to 3.7 V at 20 mA g⁻¹ with a 10-s current interruption every 5 min, as in the previous cycles. Uniform stack pressure on the cell was ensured by fixing it between two beryllium disks. Patterns, each of which took 15 min, were recorded continuously by a STOE STADI P diffractometer in transmission setup with monochromatic Cu-K$_{\alpha1}$ radiation using a Dectris Mython2 1 K detector setup. Rietveld refinements[38,39] were performed against the XRD data using the Topas-Academic software (V6)[40]. Further details of the refinements and the results are provided in Supplementary Notes 4 and 5.

## Data availability

All the experimental data used in this study are available in the Zenodo database under a Creative Commons Attribution 4.0 International License [https://doi.org/10.5281/zenodo.4964673][41].

## Code availability

The R-scripts used in the analysis of electrochemical data and the Topas refinement setting and results are available in the Zenodo database under a Creative Commons Attribution 4.0 International License [https://doi.org/10.5281/zenodo.4964673][41].

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

## Acknowledgements
The authors acknowledge STandUP for Energy consortium. H.L. thanks for the support from the projects (project numbers 42758-1, P2020-90112 and P2022-00055) funded by Swedish Energy Agency. A.S.M. is grateful to the Swedish Foundation for Strategic Research (SSF) for the financial support through the Swedish national graduate school in neutron scattering (SwedNess).

## Author contributions
Y.-C.C. co-conceived the concept, designed the experiments, performed and analyzed the electrochemical measurements, and wrote the majority of the manuscript. H.L. fabricated the electrodes and electrochemical cells, assisted with data interpretation, and revised the manuscript. A.S.M. assisted with the operando X-ray diffraction experiment, analyzed the diffraction data, and wrote about the analysis and interpretation of diffraction data in the manuscript. W.R.B. supervised the analysis and interpretation of the diffraction data and revised the manuscript. D.B. acquired the funding, supervised the project, and revised the manuscript. M.J.L. co-conceived the concept, supervised the project, and revised the manuscript.

## Funding

## Competing interests
The authors declare no competing interests.
