## [Peer Review File · Nature Communications]

REVIEWER COMMENTS

Reviewer #1 (Remarks to the Author):

Review of "A Fast Alternative to the Galvanostatic Intermittent Titration Technique"

This interesting study presents the ICI method as an alternative to GITT for estimating diffusion coefficients of lithium ions in insertion materials. ICI, which involves intermittent interruption of an applied constant current, is shown to provide similar estimates to GITT where the assumption of semi-infinite diffusion applies, and over the SOC range where kinetic limitations do not dominate. Compared to GITT, the main benefit is a significantly shorter experimental time (by 90% in the example studied). The authors exploit this to combine ICI measurements with operando XRD measurements and correlate changes in diffusion coefficient to changes in crystallographic parameters with SOC.

The paper is generally well-structured and written and the figures are well-presented. Moreover, the saving in experimental time compared to GITT makes the results significant and of interest to those working in the rapidly growing battery field in both academia and industry. The potential for combination with operando measurements is also noteworthy. In my opinion, therefore, the paper may be suitable for publication in NatureComms, with some revisions. I do, however, have some questions/concerns that I hope the authors can address.

The authors begin by demonstrating the mathematical equivalence of ICI and GITT, starting from Fick's second law and under the assumption of semi-infinite diffusion. Unfortunately, I am not able to comment in detail on the mathematical derivations which are outside of my expertise. However the validity of semi-infinite diffusion in the case of ICI (where there may be a 'background' of finite diffusion within the electrode particle at the start of each current interruption) is less intuitively obvious than for GITT (where the cell is at equilibrium at the start of each current pulse). This is not to say that the assumption is invalid - only that the paper might benefit from a short verbal justification in addition to the mathematical one.

Following the theoretical considerations, the authors demonstrate the method experimentally using 3-electrode NMC811 half cells, and show the equivalence of results obtained using ICI, GITT and also EIS.

The choice of a pouch cell was presumably to facilitate the operando XRD measurements. However, the cell design (for which the reader must refer to a previous paper) is slightly unusual, with the area of the separator being significantly larger (and a different shape) than the geometric area of the electrode. In fact the present study uses a smaller electrode area than in the reference, but does not state whether

the separator area and electrolyte volume are also different. Clarification of these details would be helpful, together with a sketch of the configuration and an explanation for why this configuration was chosen.

The unusual cell geometry prompts the question: could this affect lithium concentration gradients (and hence diffusion characteristics) in the cell? Or since $D(\text{electrolyte}) \gg D(\text{solid})$, do we assume that any concentration gradients in the electrolyte are negligible compared to those in the electrode particles? A related consideration applies to concentration gradients across the porous electrode – do we assume these are negligible? A brief mention of these points would be helpful, even if they apply equally to GITT. Similarly, the present ICI method is presumably limited to single-phase systems?

A modified GITT protocol is used to compare GITT, ICI and EIS, which is essentially a standard GITT protocol but with an EIS measurement at the end of each (1 hour) voltage relaxation period. ICI analysis is done on the voltage transient at the start of this relaxation period, after the (10 minute) GITT current pulse is switched off. In other words, the cell starts from (pseudo) equilibrium (after 1 hour rest), a 10 minute current pulse is applied and then interrupted, and ICI analysis is done on the resulting voltage relaxation. This is different to the 'standard' ICI protocol, where a constant current is applied to the cell and interrupted for 10 seconds every 5 or 15 minutes. A 10 minute current pulse may well induce finite diffusion (indeed this is stated as the reason for the downward curvature in the voltage profiles in fig. S1) but presumably to a lesser extent than in the 'standard' ICI case, where concentration gradients will essentially build up over 10 hours (assuming a C/10 rate, and assuming the effect of the intermittent 10 second interruptions is relatively small). It would be worth to acknowledge this and discuss possible effects on the results, if any.

Similarly, it would be helpful to include the results from cycle 1 (as shown in fig. 4), in fig. 6. However, a difficulty in this regard is presumably that (as noted in the figure caption), the potentials in fig. 6 are obtained during cycling (i.e. they include an overpotential) whilst the potentials in fig. 4 are OCP. Would it be possible to present an iR -corrected version of fig. 6 (and include the results from cycle 1)? Or to present D as a function of degree of lithiation, rather voltage? This would also be more useful for model parameterisation purposes. (D as a function of 'dynamic' potential is not particularly useful, since this 'dynamic' potential depends on the applied current.)

However, perhaps my main concern is that, in the ICI analysis, R and k are acquired through the linear regression of ΔE against $\sqrt{\Delta t}$, which is very similar to the procedure in GITT. It's not clear to me why the time interval used for ICI analysis (0.2-5 s after the current switch off) is so different from the interval for GITT analysis (5-40 s or 50-150 s after current switch on, depending on SOC). Figure S1 clearly shows a transition region below 5 s during the GITT current pulses, which is well-known and reflects double layer charging. A similar transition region should apply when the current is switched off. (A figure, similar to fig. S1, showing some voltage relaxations analysed using the ICI method, would be useful here.) It

therefore seems possible that the 0.2-5 s interval used for the ICI analysis would fall within this transition period. I would suggest that this should be addressed in the paper.

Some further questions/suggestions:

What are the errors bars in the figures based on? – please state this in the figure captions.

p.2 “such a test protocol makes it difficult to couple GITT with simultaneous materials characterization” – please elaborate / give references.

p.5 Eq. 14 is said to follow from eq. 13 “if the change in concentration is small”. Presumably this is the change in concentration at the particle surface during the current interruption (due to diffusion into the particle interior), but it would be helpful to clarify this.

p.8 (Experimental Methods). It is stated that the operando XRD measurements are performed every 15 minutes, while the ICI ‘measurements’, which take 10 seconds, are every 5 or 15 minutes. How long does each XRD measurement take? Is it the case that sometimes they are taken whilst the C/10 current is applied and sometimes during a current interrupt period? Would this matter?

p.9 “For the ICI method, the interval was chosen to be 0.2–5 s for the same criteria applied on the data during the rest periods”. It’s not clear what “for the same criteria applied on the data during the rest periods” means – please rephrase.

p.10 “Between 3.7 and 3.8 V, a mismatch between the GITT and ICI methods can be observed during charging, but not during discharging. The difference can be an effect of the varying charge transfer resistance and different data selection time intervals of the two techniques.” Do the authors mean varying charge transfer resistance in the charge vs. discharge direction? If so, please give references. I agree that the different time intervals may well have an effect – please see previous comments on the time interval used for ICI.

Figure 3 – are both GITT and ICI data from cycle 1? If so, can we be sure that the iR-corrected pseudo-OCP from ICI analysis in this case is representative of the result that would be obtained in the standard ICI protocol case? Also, further detail about how the iR-corrected pseudo-OCP is obtained would be useful (how is the iR drop calculated?).

p.11 “The deviation at low SoC is presumably linked to the high resistance discussed above, which interferes with the resistance determination of the ICI method.” It’s not clear to me why a high value of resistance would make it more difficult to determine that resistance – please elaborate.

Supporting Information, section 2 “Therefore, the limit of Δt increases over the number of measurements in the same course of charge or discharge since, τ_1 can be effectively accumulated due to the transient current pauses.” As noted, I am not able to verify the mathematical derivations; however, this statement seems counter-intuitive (see previous comment about a possible ‘background’ of finite diffusion).

Fig S1a and b – the linear regions appear to be 5-40 s^{0.5} and 50-150 s^{0.5} (i.e. 25-1600 s and 2500-22500 s) rather than 5-40 s and 50-150 s as stated in the text.

I am unable to comment in detail on the XRD analysis in SI section 7; however it is clearly written and the figures are well-presented.

Reviewer #2 (Remarks to the Author):

This manuscript proposes an intermittent current interrupt (ICI) method for extracting diffusion coefficients using a specific cell discharge profile. The method proposes to be faster than the traditional GITT technique and also allows easier insertion of other analysis techniques, such as XRD.

While the ICI technique itself is potentially useful, it is not particularly novel, especially to warrant publication in Nature Communications. There are plenty of papers (including by the present authors) that extract diffusion coefficients during the open circuit portion of a current interrupt (including with traditional GITT experiments). Plus, unless you have a good model of the diffusion already, a number of questionable assumptions are required about the concentration profile.

General comments:

1. If refs 8-13 already show the ICI method applied to battery systems, what is the novelty of this specific manuscript that warrants publication in this venue?

2. You focus purely on Fickian diffusion, even deriving the GITT and ICI methods assuming Fickian diffusion. Yet, non-ideal diffusion models have been gaining popularity in the modeling literature, both at the macroscale and mesoscale, showing it often represents the data more accurately. There have even been recent works using non-ideal diffusion models to extract diffusion coefficients from GITT data. Can you show how your ICI approach would work with higher-fidelity non-ideal diffusion models?

3. Your derivation of GITT assumes one method of extracting diffusion coefficients from the data. Yet there are many other methods (including your ref. 4, plus a couple of 2021 papers) of extracting GITT data, some of which are more general. If you're going to go through the trouble of deriving the GITT formulation (which I don't believe is necessary), at least mentioning the other approaches is likely warranted.

4. Doesn't the ICI approach require that the diffusion profile before the current is switched off be at a pseudo-steady state? Equation 10 has $C_0 + F(t)$ as the constant current solution. I expect this is not often the case in batteries – this could be verified through solution of a P2D-type model. In most published studies, at constant current there is a gradient of concentration within a particle (hence why we care about diffusion coefficients).

5. In Figure 1, what you're calling the ICI (compared to GITT) is really just analyzing a different part of the GITT pulse. This has already been studied in other works (papers cited in previous comment).

6. You have not directly shown the appeal of ICI – long current pulses with short interrupts.

Specific comments:

7. The title of the paper is about GITT, yet you are proposing a new method. Including your method name in the title would be more appropriate.

8. Equation quality is very poor in the PDF I reviewed.

9. Figures are not clear and are not publication quality.

10. L31: Li+ or Li?

11. L36: How long is “long enough” for the semi-infinite assumption to hold?

12. L129: Revisit this sentence – aren't you saying the same thing twice?

13. Eq 8-9: This could be simplified.

14. Paper does not seem to conform to the standard Nature Communications formatting, with methods at the end of the paper.

Reply to reviewers' comments

We would like to thank the reviewers for the detailed comments and identification of points where further clarification was required in the manuscript.

The reviewers' comments are shown in black below while our reply is typed in blue and the paragraphs inserted in the manuscript are highlighted there and repeated here in *italic*. In addition to the revisions based on the reviewers' comments, we also shifted the Methods section to the end of the manuscript and added some paragraphs in the beginning of the Results and Discussions section for the information from the Methods section that should be mentioned first.

REVIEWER COMMENTS

Reviewer #1 (Remarks to the Author):

Review of "A Fast Alternative to the Galvanostatic Intermittent Titration Technique"

This interesting study presents the ICI method as an alternative to GITT for estimating diffusion coefficients of lithium ions in insertion materials. ICI, which involves intermittent interruption of an applied constant current, is shown to provide similar estimates to GITT where the assumption of semi-infinite diffusion applies, and over the SOC range where kinetic limitations do not dominate. Compared to GITT, the main benefit is a significantly shorter experimental time (by 90% in the example studied). The authors exploit this to combine ICI measurements with operando XRD measurements and correlate changes in diffusion coefficient to changes in crystallographic parameters with SOC.

The paper is generally well-structured and written and the figures are well-presented. Moreover, the saving in experimental time compared to GITT makes the results significant and of interest to those working in the rapidly growing battery field in both academia and industry. The potential for combination with operando measurements is also noteworthy. In my opinion, therefore, the paper may be suitable for publication in NatureComms, with some revisions. I do, however, have some questions/concerns that I hope the authors can address.

We thank the reviewers for the above positive comments on the manuscript!

The authors begin by demonstrating the mathematical equivalence of ICI and GITT, starting from Fick's second law and under the assumption of semi-infinite diffusion. Unfortunately, I am not able to comment in detail on the mathematical derivations which are outside of my expertise. However the validity of semi-infinite diffusion in the case of ICI (where there may be a 'background' of finite diffusion within the electrode particle at the start of each current interruption) is less intuitively obvious than for GITT (where the cell is at equilibrium at the start of each current pulse). This is not to say that the assumption is invalid - only that the paper might benefit from a short verbal justification in addition to the mathematical one.

We thank the reviewer for pointing the lack of an intuitive interpretation of the theoretical derivation of the ICI method. Therefore, we inserted the following paragraph into the Methods section.

The above relationship between C_s and Δt is analogous to Equation 7 because the current pause can be interpreted as the result of applying a current pulse of the same magnitude but in the opposite direction on top of the existing constant current. Inserting $-I$ as I into Equation 7 cancels out the negative sign on the right side of the equation and renders Equation 15.

Following the theoretical considerations, the authors demonstrate the method experimentally using 3-electrode NMC811 half cells, and show the equivalence of results obtained using ICI, GITT and also EIS.

The choice of a pouch cell was presumably to facilitate the operando XRD measurements. However, the cell design (for which the reader must refer to a previous paper) is slightly unusual, with the area of the separator being significantly larger (and a different shape) than the geometric area of the electrode. In fact the present study uses a smaller electrode area than in the reference, but does not state whether the separator area and electrolyte volume are also different. Clarification of these details would be helpful, together with a sketch of the configuration and an explanation for why this configuration was chosen.

We thank the reviewer for pointing out this confusion in the cell format. We have now added Figure 9, which is a sketch of the cell setup and specified the volume of the electrolyte by editing the following sentence in the Methods section.

The cells were assembled in an Ar-filled glove box with 105 μ L electrolyte of 1 M LiPF₆ in a mixture of ethylene carbonate/diethylene carbonate (EC/DEC 1:1 by volume, Solvionic, purity: 99.9%) added before sealed in pouch bag material.

The unusual cell geometry prompts the question: could this affect lithium concentration gradients (and hence diffusion characteristics) in the cell? Or since $D(\text{electrolyte}) \gg D(\text{solid})$, do we assume that any concentration gradients in the electrolyte are negligible compared to those in the electrode particles? A related consideration applies to concentration gradients across the porous electrode – do we assume these are negligible? A brief mention of these points would be helpful, even if they apply equally to GITT. Similarly, the present ICI method is presumably limited to single-phase systems?

We thank the reviewer for reminding us that these points were unclear in the manuscript and we agree with the underlying assumptions mentioned by the reviewer. Therefore, we added the following two paragraphs in the Methods sections. The first is inserted right after the theoretical derivation and the second is amongst the experimental setup.

Since the ICI method is derived from the same theoretical origin as that of the GITT, the assumptions of the GITT are inherited by the ICI method, such as that the solid state diffusion in the probed electrode dominates the time-dependent cell resistance and that the material is a single-phase solid solution.

The low current density is chosen to minimize the concentration gradient in the electrolyte so that the diffusion resistance revealed by both the GITT and the ICI method is dominated by solid state diffusion.

A modified GITT protocol is used to compare GITT, ICI and EIS, which is essentially a standard GITT protocol but with an EIS measurement at the end of each (1 hour) voltage relaxation period. ICI analysis is done on the voltage transient at the start of this relaxation period, after the (10 minute) GITT current pulse is switched off. In other words, the cell starts from (pseudo) equilibrium (after 1 hour rest), a 10 minute current pulse is applied and then interrupted, and ICI analysis is done on the resulting voltage relaxation. This is different to the 'standard' ICI protocol, where a constant current is applied to the cell and interrupted for 10 seconds every 5 or 15 minutes. A 10 minute current pulse may well induce finite diffusion (indeed this is stated as the reason for the downward curvature in the voltage profiles in fig. S1) but presumably to a lesser extent than in the 'standard' ICI case, where concentration gradients will essentially build up over 10 hours (assuming a $C/10$ rate, and assuming the effect of the intermittent 10 second interruptions is relatively small). It would be worth to acknowledge this and discuss possible effects on the results, if any.

We thank the reviewer for raising the concern about effect of the duration of applying constant current before the pause for ICI analysis. We have thought about this and therefore discussed about it in the second chapter of the Supporting Information, where we address the question from the other angle and calculate the maximum time during the current pause where the ICI method is valid. Indeed, as the reviewer points out, this limit of the current pause duration is different for the semi-infinite and finite diffusion cases. Nevertheless, it can be calculated that the 5-second pause we use for the data analysis is below the limit for both cases. In other words, the duration of applying constant current before the 5-second pause for ICI analysis does not make a difference larger than 3%, which is the maximum error we allow in the calculation.

To fully address this concern from the reviewer, we plotted the ICI results from the 'modified GITT protocol' along with those from the 'standard ICI protocol' in the revised Figure 6, as the reviewer suggested subsequently. The results from the modified GITT protocol and the first cycle of the standard ICI protocol agree with each other reasonably.

Similarly, it would be helpful to include the results from cycle 1 (as shown in fig. 4), in fig. 6. However, a difficulty in this regard is presumably that (as noted in the figure caption), the potentials in fig. 6 are obtained during cycling (i.e. they include an overpotential) whilst the potentials in fig. 4 are OCP. Would it be possible to present an iR -corrected version of fig. 6 (and include the results from cycle 1)? Or to present D as a function of degree of lithiation, rather voltage? This would also be more useful for model parameterisation purposes. (D as a function of 'dynamic' potential is not particularly useful, since this 'dynamic' potential depends on the applied current.)

We agree with the reviewer that the electrode potential, which depends on the applied current, is not the most ideal choice for the x-axis. Thus, we have change the x-axis of Figure 6 to the specific capacity, which may be simply converted to degree of lithiation if needed.

However, perhaps my main concern is that, in the ICI analysis, R and k are acquired through the linear regression of ΔE against $\sqrt{\Delta t}$, which is very similar to the procedure in GITT. It's not clear to me why the time interval used for ICI analysis (0.2-5 s after the current switch off) is so different from the interval for GITT analysis (5-40 s or 50-150 s after current switch on, depending on SOC). Figure S1 clearly shows a transition region below 5 s during the GITT current pulses, which is well-known and reflects double layer charging. A similar transition region should apply when the current is switched off. (A figure, similar to fig. S1, showing some voltage relaxations analysed using the ICI method, would be useful here.) It therefore seems possible that the 0.2-5 s interval used for the ICI analysis would fall within this transition period. I would suggest that this should be addressed in the paper.

We thank the reviewer for pointing out this doubt that may be shared with other readers. Actually, in Figure S1c, the voltage relaxation analysis by the ICI method is shown alongside the GITT analyses in Figure S1a and S1b. It can be observed that the transition region, which is widely attributed to the electrochemical double layer in the GITT analysis, is not observable in Figure S1c. By applying the same principle of finding the linear part on the $E-\sqrt{t}$ plot, the interval of 0.2 to 5 seconds is chosen for the ICI method, which fortunately lies below the maximum pause duration limit determined in part 1 of the SI.

The theoretical derivations of both the GITT and the ICI method focus on the diffusion-controlled regime, leaving the solution and charge-transfer resistance as a lumped iR -drop. Therefore, although the derivation shows that the voltage change due to solid-state diffusion should be symmetric during current pulse (GITT) and pause (ICI) when the assumption of semi-infinite diffusion applies, it is not certain if the time-dependence of the charge-transfer processes would be symmetric, too. In fact, in a DFN-model based study on the GITT, published by us in collaboration with researchers at Chalmers University and since the submission of the present manuscript (<https://doi.org/10.1016/j.electacta.2021.139727>), different time-dependence of the charge-transfer overpotential was shown between current pulse and pause.

Finally, the time constant of the charge-transfer processes can also be examined by the Bode plot below, which shows the impedance spectra of cell 1 in the first cycle of the 'modified GITT protocol'. It can be seen that the second peak from the right, which should correspond to the charge-transfer process on the working electrode, is above 1 Hz for the most states of charge. This indicates that the analysis interval choices for the GITT and ICI method should both avoid the charge-transfer processes. However, the selection of the time interval may still need to reply on the inspection of the $E-\sqrt{t}$ plot and could be different for current pulses and pauses.

Some further questions/suggestions:

What are the errors bars in the figures based on? – please state this in the figure captions.

We appreciate the reviewer for reminding us of missing this vital information. The error are based on the standard deviations from the linear regression, which is now stated in the captions.

p.2 “such a test protocol makes it difficult to couple GITT with simultaneous materials characterization” – please elaborate / give references.

We thank the reviewer for pointing out the lack of clarity of this sentence. We actually would like to refer to the materials characterizations that are under strict time limit, such as the ones at a large facility. We have amended the sentence as follows.

In addition, such a test protocol makes it difficult to couple GITT with simultaneous materials characterization with time constraints, e.g. diffraction or spectroscopy at synchrotrons and neutron sources, which can provide valuable structural and/or chemical information at the moment where the process under investigation takes place.

p.5 Eq. 14 is said to follow from eq. 13 “if the change in concentration is small”. Presumably this is the change in concentration at the particle surface during the current interruption (due to diffusion into the particle interior), but it would be helpful to clarify this.

We are grateful to the reviewer for pointing these missing words out. We have now inserted “at the particle surface” after ““if the change in concentration” where we state this assumption.

p.8 (Experimental Methods). It is stated that the operando XRD measurements are performed every 15 minutes, while the ICI ‘measurements’, which take 10 seconds, are every 5 or 15 minutes. How long does each XRD measurement take? Is it the case that sometimes they are taken whilst the C/10 current is applied and sometimes during a current interrupt period? Would this matter?

We thank the reviewer for pointing out the lack of clarity here. The XRD patterns, each of which are collected for 15 minutes, are continuously collected. The midpoint of each collection duration is taken as the time coordinate of each pattern. Thus, the XRD is measured when the current is applied and interrupted shortly, but we do not think the current interruption would influence the crystallographic properties.

To avoid the confusion, we revised the sentence in the Methods section as follows.

Patterns, each of which took 15 minutes, were recorded continuously by a STOE STADI P diffractometer in transmission setup with monochromatic Cu-K α 1 radiation using a Dectris Mythen2 1K detector setup.

p.9 “For the ICI method, the interval was chosen to be 0.2–5 s for the same criteria applied on the data during the rest periods”. It’s not clear what “for the same criteria applied on the data during the rest periods” means – please rephrase.

We are sorry for the unclear sentence here, which may also lead to the above question that the reviewer raised about how the interval for the ICI method was determined. By “the same criteria”, we mean that the linear region of the $E-\sqrt{t}$ plot during current interruption was chosen for the ICI analysis, as we did for the GITT during current pulse. Examples of both can be found in Figure S3. We hope the following revised sentence would convey our message clearly.

For the ICI method, the interval was chosen to be 0.2–5 s because it contains the linear region of the $E - \sqrt{t}$ plot during current pauses.

p.10 “Between 3.7 and 3.8 V, a mismatch between the GITT and ICI methods can be observed during charging, but not during discharging. The difference can be an effect of the varying charge transfer resistance and different data selection time intervals of the two techniques.” Do the authors mean varying charge transfer resistance in the charge vs. discharge direction? If so, please give references. I agree that the different time intervals may well have an effect – please see previous comments on the time interval used for ICI.

We thank the reviewer for pointing out the lack of reference here. It has been reported in several previous studies that the GITT results from charging and discharging are different, which we have included as references in the revised sentence.

The difference can be an effect of the varying impedance measured in the course of charging and discharging^{5,24} and different data selection time intervals of the two techniques.

Figure 3 – are both GITT and ICI data from cycle 1? If so, can we be sure that the iR-corrected pseudo-OCP from ICI analysis in this case is representative of the result that would be obtained in the standard ICI protocol case? Also, further detail about how the iR-corrected pseudo-OCP is obtained would be useful (how is the iR drop calculated?).

Yes, both the GITT and ICI data are from cycle one of the ‘modified GITT protocol’ and the ICI method is conducted in the same way no matter where the data is from. We thank the reviewer for pointing out the confusion about how the iR drop is extracted and have inserted the following sentence in the introduction to the ICI method.

R and k are termed internal resistance and diffusion resistance coefficient, respectively, which are derived by extracting the intercept and slope by linear regression of ΔE against $\sqrt{\Delta t}$ automatically with a script in the R-programming language.

p.11 “The deviation at low SoC is presumably linked to the high resistance discussed above, which interferes with the resistance determination of the ICI method.” It’s not clear to me why a high value of resistance would make it more difficult to determine that resistance – please elaborate.

We appreciate the reviewer for showing the lack of clarity here. The higher charge-transfer resistance increases the time constant of the double-layer charging process, as shown by the impedance spectra in Figure S4. The larger time-constant means the potential response during the time interval where ICI analysis is applied is no longer dominated by the diffusion process. To make it clear, we revised the sentence as follows.

The deviation at low SoC is presumably linked to the high charge transfer resistance discussed above, which increases the time-constant of electrochemical double-layer charging and thus interferes with the resistance determination of the ICI method.

Supporting Information, section 2 “Therefore, the limit of Δt increases over the number of measurements in the same course of charge or discharge since, τ_1 can be effectively accumulated due to the transient current pauses.” As noted, I am not able to verify the mathematical derivations; however, this statement seems counter-intuitive (see previous comment about a possible ‘background’ of finite diffusion).

We appreciate the reviewer for pointing out the probable confusion here. In section 2 of the SI, we would like to show that the assumption we made in Equation 14 (previously Equation 12), $F(\tau_1 + \Delta t) \approx F(\tau_1)$, is valid within an accepted error during the time interval chosen for the ICI analysis. As shown in section 1 of the SI, after a constant current is applied long enough, the system leaves the semi-infinite diffusion regime and will eventually be governed by the other equation shown in Equation S5. In section 2, we show that the assumption are valid in both cases. (The case in between the two cases in Equation S5 should lie somewhere in between them.)

Perhaps the reviewer’s previous suggestion for including an intuitive explanation of the derivation of potential response during current interruption can also help with the explanation here. As described above and now in the revised manuscript, the current pause can be seen as applying a current of the same magnitude in the opposite direction of the existing constant current. Therefore, the effect introduced by the new negative current can be viewed as that caused by a new current pulse in the opposite direction, as long as the “background” electrode potential stays the same, which is what we discussed about in section 2 of the SI.

Fig S1a and b – the linear regions appear to be 5-40 s^{0.5} and 50-150 s^{0.5} (i.e. 25-1600 s and 2500-22500 s) rather than 5-40 s and 50-150 s as stated in the text.

We are grateful to the reviewer for pointing out the mistake in the label of the x-axis. It should be t instead of $t^{0.5}$. The plots are now corrected.

I am unable to comment in detail on the XRD analysis in SI section 7; however it is clearly written and the figures are well-presented.

We thank the reviewer for the positive comment.

Reviewer #2 (Remarks to the Author):

This manuscript proposes an intermittent current interrupt (ICI) method for extracting diffusion coefficients using a specific cell discharge profile. The method proposes to be faster than the traditional GITT technique and also allows easier insertion of other analysis techniques, such as XRD.

While the ICI technique itself is potentially useful, it is not particularly novel, especially to warrant publication in Nature Communications. There are plenty of papers (including by the present authors) that extract diffusion coefficients during the open circuit portion of a current interrupt (including with traditional GITT experiments). Plus, unless you have a good model of the diffusion already, a number of questionable assumptions are required about the concentration profile.

General comments:

1. If refs 8-13 already show the ICI method applied to battery systems, what is the novelty of this specific manuscript that warrants publication in this venue?

We thank the reviewer for raising this concern for the novelty. In the previous studies, refs 8–13, the ICI method is used for extracting the time-independent and time-dependent resistances. Here, the method is further developed for the derivation of diffusion coefficient. We have inserted the following to make this point clear in the manuscript.

The intermittent current interruption (ICI) method was originally designed for continuous resistance measurements. During constant-current cycling of a diffusion-controlled system, the method introduces transient current pauses, in which the change in electrode potential (ΔE) and time (Δt) since the current (I) is switched off can be expressed as follows.^{9,10,19}

$$\Delta E(\Delta t) = E(\Delta t) - E_i = -IR - Ik\sqrt{\Delta t} \quad 1$$

where E_i is the potential right before the current is switched off, and R and k are termed internal resistance and diffusion resistance coefficient, respectively, which are derived by extracting the intercept and slope by linear regression of ΔE against $\sqrt{\Delta t}$ automatically with a script in the R-programming language.²⁰ In this work, the method will be further developed to derive the diffusion coefficient of the charge carrier in insertion-type electrode

materials, which is else mostly done using the galvanostatic intermittent titration technique (GITT) in the literature.^{1,2,4-6,21-23}

2. You focus purely on Fickian diffusion, even deriving the GITT and ICI methods assuming Fickian diffusion. Yet, non-ideal diffusion models have been gaining popularity in the modeling literature, both at the macroscale and mesoscale, showing it often represents the data more accurately. There have even been recent works using non-ideal diffusion models to extract diffusion coefficients from GITT data. Can you show how your ICI approach would work with higher-fidelity non-ideal diffusion models?

We thank the reviewer for pointing out the limitations of only considering Fickian diffusion. As discussed in the Introduction sections, the research community has been acknowledging the possible pitfalls when using an oversimplified GITT analysis. However, as we also pointed out in the Introduction, the most practical way of ensuring the validity of GITT analysis is to select the experimental data that lie in the semi-infinite diffusion regime, which is what we discuss in a considerable portion of the text. And, hopefully, we also manage to demonstrate that built on the same framework of the theory of this improved GITT, an ICI method can be a time-efficient alternative.

Of course, it would be a great idea to implement more complicated diffusion model on the analysis of the ICI method. However, such implementation is outside the scope of this work.

3. Your derivation of GITT assumes one method of extracting diffusion coefficients from the data. Yet there are many other methods (including your ref. 4, plus a couple of 2021 papers) of extracting GITT data, some of which are more general. If you're going to go through the trouble of deriving the GITT formulation (which I don't believe is necessary), at least mentioning the other approaches is likely warranted.

We thank the reviewer for expressing the concern for insufficient referencing and too many details in the mathematical derivation. Indeed, as the reviewer mentioned, reference 4 is nice recent work on the improvement of analysis of the GITT. Therefore, we discussed about the methods used in the paper in the "Experimental execution of GITT" section and compare our methods directly to theirs. As for the latest works in 2021, we included reference 21, which is a nice representation of the works on similar topics in our opinion.

4. Doesn't the ICI approach require that the diffusion profile before the current is switched off be at a pseudo-steady state? Equation 10 has $C_0 + F(t)$ as the constant current solution. I expect this is not often the case in batteries – this could be verified through solution of a P2D-type model. In most published studies, at constant current there is a gradient of concentration within a particle (hence why we care about diffusion coefficients).

We appreciate the reviewer for pointing out the importance of reaching a pseudo-steady state. This is the reason why we elaborated about the limitation on the current interruption duration in section 2 of the SI, which is indeed dependent on the duration of the applied constant current before the interruption. We also demonstrated that in our choice of time interval for data analysis, the assumption of $F(t_1 + \Delta t) \approx F(t_1)$ in Equation 14 (previously Equation 10) is valid.

5. In Figure 1, what you're calling the ICI (compared to GITT) is really just analyzing a different part of the GITT pulse. This has already been studied in other works (papers cited in previous comment).

Indeed, as the reviewer stated, there has been previous work on analyzing the rest period of the GITT protocol. However, the novelty of the method proposed here is minimizing the rest period and thus the disruption to the constant current cycling, while achieving the same goal, deriving the diffusion coefficient. As Reviewer 1 pointed out, the experimental validation of the ICI method is done through a modified GITT protocol. Therefore, we hope that the comparison of the ICI data from the 'modified GITT protocol' and 'standard ICI protocol' in the new version of Figure 6 can also answer Reviewer 2's question here.

6. You have not directly shown the appeal of ICI – long current pulses with short interrupts.

We are grateful to the reviewer for pointing out this important aspect. Thanks to the movement of the Methods section to the end of the manuscript, we take the opportunity of inserting a bridging paragraph to emphasize the novelty of the ICI method. The parts related to the appeal of the ICI method is as follows.

The applications of both the GITT and the ICI method are illustrated by parts of the raw data of the same cell in different cycles in Figure 1. On the working electrode of $\text{LiNi}_{0.8}\text{Mn}_{0.1}\text{Co}_{0.1}\text{O}_2$ (NMC811), the ICI method introduces short pauses (10 s every 300 s) during constant-current ($C/10$, $C = 200 \text{ mA g}^{-1}$) charging while the GITT applies short current pulses ($C/10$, 600 s) in between long rests (>1 h) for reaching the open-circuit potential (OCP). Therefore, the ICI method probes the same range of states of charge in less than 15% of the time required by the GITT.

Specific comments:

7. The title of the paper is about GITT, yet you are proposing a new method. Including your method name in the title would be more appropriate.

We thank the reviewer for the suggestion. The balance between insufficient information and a concise title is always hard to strike. We believe that the message of the current title is one of the essential summary of the manuscript: the ICI method delivers what the GITT does in a more efficient way. Nevertheless, we are open to new titles if this is allowed by the editorial office.

8. Equation quality is very poor in the PDF I reviewed.

We apologize for the low quality of the PDF file. We think this may be caused by the conversion of files automatically done in the system since we uploaded a word file. We will upload a PDF file of the revised manuscript to avoid this issue.

9. Figures are not clear and are not publication quality.

We apologize for this again and hope this will not happen for the revision.

10. L31: Li⁺ or Li?

We think Li⁺ would be more appropriate here since sentence refers to the charge carrier. 'Li' would suggest that Li in the electrode material is charge-neutral, which would be contradictory to the fact that its concentration changes the electrical potential?

11. L36: How long is "long enough" for the semi-infinite assumption to hold?

We thank the reviewer for raising this question but the exact answer to this depends on the diffusion coefficient and dimensions of the electrode, which is discussed in details in section 3 of the SI.

12. L129: Revisit this sentence – aren't you saying the same thing twice?

We thank the reviewer for pointing this out. We have streamlined the paragraph in the new version of the manuscript.

13. Eq 8-9: This could be simplified.

We thank the reviewer for informing us but we are not very experienced with mathematical expressions and thus cannot spot this redundancy. We hope this could be simply revised in the upcoming communication between us.

14. Paper does not seem to conform to the standard Nature Communications formatting, with methods at the end of the paper.

We thank the reviewer for raising this issue. The revised manuscript has the Methods section in the end.

REVIEWER COMMENTS

Reviewer #1 (Remarks to the Author):

I am mostly content that the revised manuscript addresses my previous comments and am happy to recommend it for publication in Nature Comms, except for the following:

- references in the text to figure numbers from figure 2 onwards are no longer correct
- regarding the different time intervals used for ICI and GITT analysis, I appreciate the authors' comments around asymmetry in the time dependence of charge transfer (could this be referenced in the article/SI?). However they show via a Bode plot that the charge transfer process lies between 1-10 Hz. Doesn't this overlap with the time interval used for ICI analysis (0.2-5 s)? What happens if an interval of 1-5 s is used instead? Also, as a minor point, could fig S1c be referenced in the text somewhere (line 163 of the article perhaps) as otherwise it's easy to miss (as I did!) that this figure is for ICI rather than GITT.

Reviewer #2 (Remarks to the Author):

I appreciate the responses by the authors. While I am not completely satisfied with all of the responses, I have no additional comments at this time.

Response to reviewers

Reviewer #1 (R1)

R1: I am mostly content that the revised manuscript addresses my previous comments and am happy to recommend it for publication in Nature Comms, except for the following:

- references in the text to figure numbers from figure 2 onwards are no longer correct

A: We apologise for these errors, and have corrected these in the revised version.

R1: regarding the different time intervals used for ICI and GITT analysis, I appreciate the authors' comments around asymmetry in the time dependence of charge transfer (could this be referenced in the article/SI?). However they show via a Bode plot that the charge transfer process lies between 1-10 Hz. Doesn't this overlap with the time interval used for ICI analysis (0.2-5 s)? What happens if an interval of 1-5 s is used instead? Also, as a minor point, could fig S1c be referenced in the text somewhere (line 163 of the article perhaps) as otherwise it's easy to miss (as I did!) that this figure is for ICI rather than GITT.

A: Indeed, we have now changed the time interval for the ICI analysis to 1–5 seconds and found very similar results. Therefore, all the ICI results are now presented with the new time interval thanks to the reviewer's suggestion. We used 0.2–5 seconds in the first place by identifying the linear region on the $E-\sqrt{t}$ plot, but now with a closer look at the Bode plot as requested by the reviewer, it does make more sense to use the interval 1–5 seconds. We thank the reviewer for the helpful advice!

As for Figure S1c, it is now mentioned at line 163 of the previous version of revised manuscript. (line 170 in the latest version)

Reviewer #2 (R2)

R2: I appreciate the responses by the authors. While I am not completely satisfied with all of the responses, I have no additional comments at this time.

A: We regret that the reviewer was not completely satisfied with our previous responses, and so have made efforts in this revision to further address what we believe to be the main points of concern raised by the reviewer in the previous round of review.

Firstly, we recognise that the importance of the assumption of Fickian diffusion should be discussed and placed in the context of research into non-Fickian diffusion models. We wish to stress here that the intended scope of this manuscript was to investigate and demonstrate the applicability of determining the diffusion coefficient from a short current interruption based on the same principles as the GITT method proposed by Weppner and Huggins, which assumes Fick's law diffusion. Two major motivations for this assumption are that this version of the GITT method is widely known and applied in the field, and most models of the electrochemistry of Li-ion batteries used both in the literature and in commercial products make the same assumption. A second consideration is the complexity of incorporating the model into the analytical method; some higher fidelity models which consider non-ideal diffusion (e.g. as proposed in a recent paper by Horner et al, doi: 10.1021/acsaem.1c02218) require the solving of differential equations where there is no analytical solution, which would be a substantial undertaking which we could not reproduce and evaluate in the

scope of this study. Despite this, we do not believe there is any intrinsic obstacle to extending the ICI method to other models of diffusion behaviour in future work, and we release our raw data and code publicly as part of this manuscript to encourage just this. With this in mind, we have made the following edits in our revised version:

- We have revised the introduction of our manuscript to clarify the intended scope, as outlined above
- We have added a discussion of the limitations of the study in which we address the fundamental assumptions and limitations of GITT and ICI and the scope for future work incorporating non-Fickian diffusion models.
- We have also revised the title and abstract to strengthen the intended scope of this manuscript accordingly.

Secondly, we recognise that concern may remain over the validation of the method and its assumptions regarding the choice of time intervals. We recognise that the reviewer raised concerns about key assumptions in their point #4 in the previous round of review. With consideration of the fundamental assumptions covered above, we are confident that the other assumptions made to adapt the analytical approach to the ICI method are reasonable and the different methods evaluated give consistent results. With this revision we have sought to strengthen this with a focus on the following aspects.

Reviewer 2's previous comment claiming that a requirement of the ICI method is that the diffusion profile should be at a pseudo-steady state was not completely clear to us. Therefore, we feel that our previous response may not fully address the comment.

First, there could be a confusion over the definition of "pseudo steady state". To our understanding, pseudo-steady state refers to a state where the concentration gradient in the spatial coordinates is time-invariant but the concentration changes, i.e. $\Delta c/\Delta t$ is constant across the spatial coordinates. However, upon revisiting Reviewer 2's previous comment 4, it seems like the Reviewer referred to a state where the concentration gradient is absent, which we think would be more commonly denoted as the "steady state". There should be a concentration gradient in the NMC particle under constant current charging/discharging and the ICI method makes use of the disappearance of this gradient when the current is stopped to derive the diffusion coefficient.

Second, the derivation of the ICI method actually does not require the system to be at either steady state or pseudo steady state at the moment when the current is interrupted. It only assumes the following is true in Equation 14:

$$F(\tau_1 + \Delta t) \approx F(\tau_1)$$

to simplify the expression for the surface concentration. To examine the validity of this assumption, we look into different diffusion scenarios at the moment where the current is interrupted in Section 2 of the SI. Under the conditions used in this manuscript, namely C/10 with 5 s interruptions at 5 min intervals, we have in principle 120 interruptions per half-cycle; by the second interruption, the ratio $F(\tau_1)/F(\tau_1 + \Delta t)$ is already >99%, which we consider is sufficient justification to make this assumption. We have clarified this in the revision:

During constant-current cycling of a diffusion-controlled system, the method introduces transient current pauses, in which the change in electrode potential (ΔE) and time (Δt) since the current (I) is switched off can be expressed as Equation 1,^{10,11,20} given Δt is sufficiently small, as elaborated in section 2 of the SI.

Additionally, we wish to highlight the counterpart article to this manuscript, which was submitted after this manuscript but published during the revision process of this manuscript, in which we

validated this method alongside GITT and EIS using a P2D model (doi: 10.1016/j.electacta.2021.139727). This study demonstrated that the above assumption does not significantly contribute to the experimental error, which is instead more strongly affected by other factors, similar to GITT. The factors studied included the determination of the change in potential with respect to concentration (dE/dC), as well as the fitting duration, thickness of the electrodes, and applied current. Together with the experimental validation in the present manuscript, we believe this comprises a thorough discussion of the factors affecting the estimation or determination of D , which must be fully considered and accounted for regardless of the diffusion model or experimental method.